# “When the Word Is Too Big, It’s Just Too Hard”: Stroke Survivors’ Perspectives About Health Literacy and Delivery of Health Information [note 1]

**DOI:** 10.3390/healthcare13050541

**Published:** 2025-03-03

**Authors:** Dana Wong, Lauren M. Sanders, Alison Beauchamp, Claire Formby, Emma E. Smith, Creina Hansen, Kathryn McKinley, Karella De Jongh, Karen Borschmann

**Affiliations:** 1School of Psychology and Public Health, La Trobe University, Bundoora, VIC 3086, Australia; d.wong@latrobe.edu.au (D.W.); e.smith5@latrobe.edu.au (E.E.S.); 2Department of Neurosciences, St Vincent’s Hospital Melbourne, Fitzroy, VIC 3065, Australiacreina.hansen@svha.org.au (C.H.); 3Department of Medicine, Melbourne Medical School, University of Melbourne, Parkville, VIC 3010, Australia; 4School of Rural Health, Monash University, Warragul, VIC 3820, Australia; alison.beauchamp@monash.edu; 5Health Independence Program, St Vincent’s Hospital Melbourne, Fitzroy, VIC 3065, Australia; claire.formby@svha.org.au; 6School of Psychological Sciences, University of Melbourne, Parkville, VIC 3010, Australia; 7Allied Health, Northern Health, Epping, VIC 3076, Australia; kathryn.mckinley@nh.org.au; 8Speech Pathology, St Vincent’s Hospital Melbourne, Fitzroy, VIC 3065, Australia; 9Allied Health, St Vincent’s Hospital Melbourne, Fitzroy, VIC 3065, Australia; 10The Florey Institute, University of Melbourne, Heidelberg, VIC 3084, Australia

**Keywords:** stroke, health literacy, organisational literacy, Ophelia methodology, health information, cognitive impairment, communication disability, cultural and linguistic diversity

## Abstract

**Background:** Health literacy can impact comprehension, recall, and implementation of stroke-related information, especially in the context of cognitive and communication impairments, cultural-linguistic diversity, or ageing. Yet there are few published lived experience perspectives to inform tailoring of health information. **Objectives:** We aimed to (i) explore perspectives about the impact of health literacy on information needs and preferences of stroke survivors with diverse characteristics; and (ii) identify ways to better tailor information delivery for stroke survivors with low health literacy. **Methods:** This qualitative study was conducted using the Ophelia (Optimising Health Literacy and Access) methodology. First, health literacy information was collected from participants. Hierarchical cluster analysis was used to identify different health literacy profiles within the participant sample. Four profiles were identified, from which four case vignettes were created. Second, focus groups and interviews were conducted to explore the health information needs and preferences of the case vignettes. Qualitative data were analysed with reflexive thematic analysis. **Results:** Nineteen people participated (median (IQR) age = 65 (49, 69), 10 (53%) female); five used interpreters. Participants represented diverse socioeconomic, cultural, and stroke-related characteristics, and generally had low health literacy. Four qualitative themes were generated highlighting the impact of *Individual knowledge, capacity, and beliefs about stroke and health services* on people’s capacity to engage with stroke-related information; *Tailoring and personalisation of information delivery* to the patient’s knowledge, capacity, and beliefs; *Having a support network to rely on*; and patients *Feeling like I am in safe hands* of clinicians and services. **Conclusions:** Findings provide several important directions for improving accessible stroke information delivery suitable for people with all levels of health literacy, and to optimise patient understanding, recall, and implementation of healthcare information.

## 1. Introduction

Low health literacy, defined as people’s ability to seek, understand, engage with, and implement health information, is surprisingly prevalent in the general community. Only 41% of Australian adults have sufficient health literacy to understand and use health information [1,2]. Low health literacy is particularly common in people with older age, limited education, and in some culturally and linguistically diverse (CALD) populations AIHW [2,3]. One in two people with stroke experience cognitive impairment and one in four have communication impairments [4]. This can further compromise health literacy and affect comprehension, recall, and effective implementation of health-related information [5,6]. With low health literacy impacting poorer stroke outcomes including medication adherence, general health status, and hospital readmission [7], it is incumbent on health practitioners to provide appropriate health information to support stroke recovery

Organisational literacy, or the health literacy environment, is the degree to which organisations can acknowledge and accommodate variations in patient health literacy and thereby support patients to manage their health and navigate the health system. Proponents of improving organisational literacy, rather than attempting to improve health literacy, support a “universal precautions” approach to heath communication. Such an approach requires habitual use of clear communications to improve accessibility of health information for all people with stroke, regardless of their health literacy [8]. This shifts the onus of responsibility from the marginalised patient to that of the health service to provide appropriately tailored stroke care [9]. Health services have a responsibility to support health literacy [10]; however, to date there is limited evidence that this occurs consistently in clinical practice. A recent qualitative study of Australian stroke clinicians found that only 60% had received training to support their communication with people with aphasia [11], and there are gaps between clinicians’ theoretical understanding of information provision and their actual practice [12]. Although there are lived experience-informed guidelines for the provision of information for people with stroke [13], adult learning principles are not always applied by health professionals when providing information [14], and the reading level of many stroke education materials is too high [15].

Poor organisational health literacy can lead to poor experiences of healthcare and poor outcomes post-stroke [2]. Therefore, it is incumbent upon stroke services to improve their awareness of and responsiveness to low health literacy in their service users. To do this, it is important to include the voices of people with lived experience of stroke to ensure that service improvements reflect the values and needs of patients. In particular, it is important to include the voices of people with cognitive and communication difficulties, limited education, and those from CALD backgrounds, given their increased likelihood of low health literacy and their frequent exclusion from stroke research [16].

Our study had two main aims. Firstly, we aimed to explore stroke survivors’ perspectives on health literacy and how it may impact the information needs and preferences of people with stroke, including people typically under-represented in stroke research. Secondly, we aimed to identify targets or directions for improving organisational literacy—i.e., ways to enable stroke services to better tailor information delivery to stroke survivors with low health literacy, and therefore support their patients to understand and use stroke information to optimise their outcomes.

## 2. Materials and Methods

This qualitative study was conducted in two stages using the Ophelia (Optimising Health Literacy and Access) methodology [17]. In Stage 1, health literacy information was collected from participants to create case vignettes that represented their common characteristics. In Stage 2, focus groups and interviews were conducted about the health information needs and preferences of the people described in the case vignettes. This article is a revised and expanded version of a paper entitled “When the Word is Too Big, it’s Just Too Hard: How can Clinicians Support Patients’ Health Literacy to Improve Recovery after Stroke?”, which was presented at Stroke 2023, Melbourne, Australia, in August 2023 [18].

### 2.1. Participants

Inclusion criteria for the study were community-dwelling adults (≥18 years old) with stroke or transient ischemic attack who attended St Vincent’s Hospital Melbourne (SVHM) outpatient stroke clinic between January 2021 and February 2022. SVHM is a public tertiary hospital in inner Melbourne that caters to a broad demographic of the local multicultural community.

Clinic lists were screened by clinician researchers (CF and LS) and consecutive, eligible participants were invited to participate via a telephone call or discussion during a stroke clinic appointment. Purposeful recruitment was undertaken to include people whose preferred language was not English (especially people from Vietnamese background as this population was a common user group of interpreter services at SVHM). Potential participants identified as likely to have aphasia, cognitive impairment, or low English literacy were offered an Assisted Communication or Easy English version of the Patient Information and Consent Form, with use of interpreters when required. Participants were made aware that this research was being conducted with aim of improving stroke services at SVHM. Further information about the research teams’ interest in the topic was discussed according to patient interest. Participant recruitment was limited by strict and prolonged COVID-19 related lockdowns in Melbourne during the data collection period.

### 2.2. Materials and Procedures

Materials were developed and pilot tested by the multidisciplinary research team, who had extensive research and clinical experience in stroke recovery, supported communication (for post-stroke aphasia and cognition difficulties), health literacy, and interpreter services, and lived experience of stroke.

#### 2.2.1. Stage 1

Electronic health records were reviewed to collect patient data relating to demographics (gender, age, languages spoken, highest level of education, birth country, living arrangements, carer support), clinical details of stroke including any description or assessment of language impairment, communication support needs and cognition, and global disability (modified Rankin score, mRS). Education level was classified as follows: did not complete primary school, completed primary school, completed secondary school, certificate/apprenticeship/diploma, degree, or post-graduate qualification. Observed communication or cognition impairments were also noted during participant interviews that were conducted by researcher CF, an experienced stroke clinician.

Socioeconomic status was categorised based on postcode of home address using the Index of Relative Socio-economic Advantage and Disadvantage, (1 = most disadvantaged, 5 = most advantaged) ABS [19]. Lower scores indicate relatively greater disadvantage and a lack of advantage in general. For example, an area could have a low score if there are many households with low incomes, or many people in unskilled occupations, and a few households with high incomes, or few people in skilled occupations.

Structured interviews, developed collaboratively by the research team, were conducted to collect information not available in medical records, and participant data relating to health literacy, global disability (modified Rankin score, mRS), stroke-related information needs, and knowledge of stroke and secondary stroke prevention. Interviews were undertaken in person, via telephone or via telehealth, depending on patient preference and COVID-19 pandemic restrictions. Interviews were conducted by author CF (an experienced stroke clinician and researcher) between December 2021 and March 2022. Interpreters were used as required. Participants were offered breaks during the interview and interviews were conducted across two sessions, if needed.

Formal cognitive assessment using the Oxford Cognitive Screen (Australian version, OCS-AU) was planned; however, due to challenges with telehealth administration during the pandemic, this was not conducted.

Health literacy was evaluated using the Health Literacy Questionnaire (HLQ) [20] and the Brief Health Literacy Screening Tool (BRIEF) [21]. The HLQ measures health literacy across nine independent scales, each measuring a different aspect of health literacy. The HLQ is considered highly reliable (composite reliability ranges from 0.8 to 0.9) [22] and is widely used, including in populations with cardiovascular disease [23]. Five HLQ scales were used for this study: Scale 2, Having sufficient information to manage my health; Scale 3, Actively managing my health; Scale 4, Social support for health; Scale 6, Ability to actively engage with healthcare providers; and Scale 7, Navigating the healthcare system. Scales 2, 3, and 4 are answered using a 4-point Likert scale (range 1–4) and scales 6 and 7 answered using a 5-point Likert scale (range 1–5). The Brief Health Literacy Screening Tool (BRIEF) [24] is a 4-item measure (range 1–20) that captures people’s functional health literacy (i.e., the ability to read and understand written information). The instrument has been widely used across different health conditions, including stroke [25].

#### 2.2.2. Stage 2

Using HLQ data collected in Stage 1, four different groupings (“clusters”) of participants were identified, each representing a different health literacy profile within the sample (see Section 2.3 for detail). Brief case vignettes were then developed representing the four participant clusters. An example vignette is contained in Table 1; the remaining three vignettes are Appendix A. The vignettes were then used to guide discussions in the Stage 2 focus groups and interviews.

Participants were invited to attend small focus groups. Focus groups were conducted in June 2022 via Zoom by DW (experienced clinical neuropsychologist, group facilitator, and qualitative researcher), AB (experienced health literacy researcher), and CF (clinician researcher), all of whom are female. Each group commenced with a brief introduction about the interviewing team and the aims of the project, which were (1) to improve the way stroke services were delivered at both SVHM and more broadly, and (2) to make it easier for survivors of stroke to understand and use information about their health. Participants were also provided with the opportunity to introduce themselves. Using PowerPoint slides and a verbal description, participants were presented with 1–2 vignettes (see Table 1) which best represented the experiences of the group. Participants were then asked a series of questions about how stroke services could meet the needs of the ”character” in the vignette. Questions included (i) “Does this sound like someone you know, or something you may have experienced?”, (ii) “What things might make it difficult for [character] to find, understand, and use information about [their] stroke?”, (iii) “What strengths does she have to help her make changes?”, and (iv) “What could our health service do to make things easier/better for [character]?” The group facilitators encouraged all participants to share their views and ensured that everyone had the chance to do so either verbally or in the Zoom chat. Communication support strategies, such as slowed pace of speech, repetition, paraphrasing, and reflective summaries of participants’ comments to confirm their meaning, were used by facilitators as required. Interviewers made notes during discussions, and a summary of participant comments was then shown on a slide for participants to confirm that the summary accurately described their views.

For participants who indicated that participation in focus groups was too challenging, individual semi-structured interviews were undertaken via Zoom or telephone by author CF with an interpreter when required. Duration ranged from 15 to 60 min. For telephone interviews, a spoken description of the vignettes was provided and similar questions to that of the focus groups described above used as prompts to elicit information.

Audio from each focus group and interview was recorded and transcribed verbatim. If the participant did not wish to be recorded, a written summary of their interview was taken.

### 2.3. Data Analysis

#### 2.3.1. Stage 1

Data related to individual participants’ demographics, health literacy, and stroke-related information needs were analysed descriptively. STATA version 15 [26] and SPSS version 22 [27] were used for analyses of quantitative data. A *p*-value of <0.05 was assumed for statistical significance. Hierarchical cluster analysis was used to identify different health literacy profiles within the patient sample using Ward’s method for linkage [28]. Based on previous work [17], a range of cluster solutions of between 2 and 8 clusters was pre-determined. Selection of the most appropriate cluster solution was based on two criteria: first, whether the standard deviation within each scale within each cluster was below 0.6; and second, whether distinct patterns of HLQ scale scores were seen between clusters. Demographic, clinical, and health data were reported for each cluster, providing a detailed picture of a “typical” person within that cluster.

#### 2.3.2. Stage 2

The focus groups and interviews were transcribed and analysed using reflexive thematic analysis [29,30]. A critical realist approach to analysis was taken, seeking to understand the meaning participants made of their experiences (via the case vignettes) and the influence of broader social and structural contexts, within the shared context of engaging with health services following a stroke. Data were coded through a process of familiarisation (rereading the transcripts several times), then generating initial codes based on both verbatim utterances and underlying patterns and concepts, and constantly revisiting the transcripts as the codes were refined. Generation and refinement of codes was conducted by researcher ES (research assistant with Honours-level training in psychology) in consultation with DW, using NVivo 1.7.1. The research team (CF, CH, DW, ES, KB) then collaboratively grouped the codes and generated themes during a Zoom meeting using Ideaflip online software (ideaflip.com). Themes were defined and labelled, and relationships between themes were explored together as a team.

## 3. Results

### 3.1. Stage 1

Nineteen participants completed structured interviews in Stage 1. Participants were interviewed via telephone (n = 16) in their homes or local communities, face to face (n = 2) in the stroke clinic or via Zoom (n = 1) in their home. Interpreters were utilised in five interviews. Family members were present for all face to face, Zoom, and interpreter interviews. It is unknown if anyone else was present during telephone calls conducted in English.

Participant characteristics are shown in Table 2. The median (IQR) age of participants was 65 (49, 69) years, 10 (53%) were female and 9 (47%) were male, and 9 (47%) completed education beyond high school. Eleven participants (58%) resided in areas of the highest socioeconomic category, and two (11%) resided in the lowest category. Eleven participants (58%) were born in Australia, and the remainder in Asia or Europe. Five participants (26%) spoke Vietnamese, and all used interpreters during their interviews. The remaining 14 people were interviewed in English without interpreters. The median time post-stroke was 9.5 months (IQR 6, 14). As mentioned, cognitive assessment could not be completed, and medical records generally had no record of cognitive status. Cognitive impairment was noted for six (32%) participants; three of these by the researcher based on clinical impression, and three by the family (clinical impression was more difficult for these participants due to use of an interpreter). Seven (37%) participants had mild communication impairments at the time of stroke (NIHSS 1–3 for aphasia or dysarthria). All participants completed the interviews independently, without carer support or personalised communication support.

HLQ data approximated normal distribution, and homogeneity of variance was not violated. Mean HLQ scores are shown in Table 3. For scales 2, 3, and 4 (maximum possible score 4.00), the lowest score was seen for scale 2, Having sufficient information to manage my health (mean score 2.67, SD 0.76). For scales 6 and 7 (maximum possible score 5.00), the lowest mean score was for scale 7, Navigating the healthcare system (mean score 3.38, SD 1.01). For the BRIEF, median score was 12 (inter-quartile range 8, 19) from a possible range of 4–20.

From the cluster analysis of HLQ data, four distinct profiles were identified, representing a diversity of health literacy strengths and weaknesses, as shown in Table 4.

### 3.2. Stage 2

Ten participants from the Stage 1 cohort participated in Stage 2. Nine people declined to participate in Stage 2 (carer responsibilities n = 1; time constraints n= 4; no longer wished to participate n= 1; new illness n = 1; unable to be contacted n = 2). Of those who participated in Stage 2, 6 (60%) were female and 4 (40%) male, median (IQR) age = 57 (38, 65); time since stroke = 10 (7, 14) months, and 7 (70%) completed education beyond high school). One participant in Stage 2 spoke Vietnamese and completed the interview with an interpreter. Four people (40%) were identified as having cognitive impairment, and three (30%) had communication impairment. The rates of cognitive and communication impairments noted in the group of people who did not complete Stage 2 were 2/9 (22%) and 4/9 (44%), respectively.

Two 90 min focus groups were undertaken, each with three participants. Four individual interviews were undertaken via telephone (n = 3) or Zoom (n = 1). Due to time constraints and the rich discussion generated by the vignettes, no participants were presented with all four vignettes. All focus group participants were presented with one vignette. Two interview participants were presented with one vignette, one interview participant was presented with two vignettes, and one participant did not wish to hear the vignette and was interviewed about their own experiences. One participant did not wish to be recorded, so instead a summary of their interview was written by the researcher. Despite the attrition between Stage 1 and 2 of the study, we deemed theoretical sufficiency [31] to be achieved after 10 interviews.

#### Reflexive Thematic Analysis Findings

As depicted in Figure 1, four themes and one subtheme were generated about health literacy needs and preferences of stroke survivors, and how cognitive difficulties, communicative difficulties, and culture may affect these needs and preferences. The first theme, “Individual knowledge, capacity, and beliefs about stroke and health services”, considers the individual characteristics, history, and worldview of the patient, and the subtheme, “Systemic and societal context influencing individual stroke literacy”, considers the influence of the patient’s context on their individual knowledge, capacity, and beliefs about stroke and health services. The second theme, “Tailoring and personalisation of information delivery”, considers the characteristics of the healthcare information presented to the patient, its delivery to the patient, and the value of tailoring information and delivery to the patient based on their knowledge, capacity, and beliefs. The third theme, “Having a support network to rely on”, considers the multifaceted roles of support people in facilitating access to healthcare information and supporting patients to implement healthcare recommendations. The final theme, “Feeling like I am in safe hands”, considers the extent to which the patient trusts and is confident in the quality of care they receive from clinicians and services. Feeling in safe hands is influenced by the patient’s knowledge, capacity, and beliefs (Theme 1), the extent to which information delivery is tailored to their needs (Theme 2), and the interactions of their support network, clinicians, and services (Theme 3). Each of these themes influences understanding, recall, and implementation of healthcare information by the patient.

*Theme 1: Individual knowledge, capacity, and beliefs about stroke and health services.* This theme pertains to the influence of individual knowledge, beliefs, strengths, and challenges on how patients make sense of their stroke journey and health information. This theme also contains a subtheme, *Systemic and societal context influencing individual stroke literacy.* Table 5 describes the key concepts reflected in Theme 1 and its subtheme, and provides illustrative quotes.

*Theme 2: Tailoring and personalisation of information delivery.* This theme highlights the importance of delivering healthcare information in a manner that is relevant and meaningful for each individual patient. There is no “one size fits all” approach, so healthcare providers need to use a range of strategies to meet the needs and preferences of individual patients. As outlined in Table 6, participants stressed the importance of tailoring information delivery to their needs, knowledge, capacity, and beliefs. When materials, resources, and information delivery are appropriate to the needs and capacity of the stroke survivor, they support the understanding, recall, and implementation of healthcare information. Specific suggestions made by participants for support materials and information delivery methods are listed in Table 7.

*Theme 3: Having a support network to rely on.* This theme identifies the importance of having access to friends, family, and services to assist in understanding, recalling, and implementing healthcare information. As shown in Table 8, supportive friends and family were crucial allies for participants at every stage of the stroke journey, from the acute stage to ongoing chronic care. They helped the participants to feel cared for, that there was someone with whom to share the difficult experiences, and that empowered them to understand and respond to health information together.

*Theme 4: Feeling like I am in safe hands.* This theme highlights the importance of having access to effective healthcare services, feeling confident in the quality of care, and feeling safe to speak up and ask questions. When patients feel they are in safe hands, they believe that their healthcare team is competent, trustworthy, and has their best interests at heart, and that they are working together in alliance. This enables participants to better engage with health information. Table 9 describes key concepts important for “feeling like I am in safe hands”.

## 4. Discussion

Using the novel Ophelia (Optimising Health Literacy and Access) methodology, the aim of this study was to explore perspectives on the associations between health literacy and the information needs and preferences of stroke survivors, and identify targets or directions for improving organisational health literacy to better tailor information delivery. Our participants, including people typically under-represented in stroke research (i.e., those from CALD backgrounds, and with cognitive and communication impairments), provided rich insights into the impact of health literacy on their ability to seek, understand, engage with, and act on health information. Four themes were generated in discussions about the four case vignettes used to describe typical health literacy profiles. The first theme highlighted the impact of *Individual knowledge, capacity, and beliefs about stroke and health services* on their capacity to engage with stroke-related information. The second theme, *Tailoring and personalisation of information delivery*, pointed to the importance of accessible healthcare information delivered in a manner that is tailored to the patient’s knowledge, capacity, and beliefs—rather than adopting a “one-size-fits-all” approach. Thirdly, *Having a support network to rely on* emphasised the multifaceted roles of family and other support people in facilitating access to, comprehension of, and implementation of healthcare information. Finally, *Feeling like I am in safe hands* described the importance of patient trust and confidence in the quality of care they receive from clinicians and services. Our findings provide several important directions for improving organisational literacy to optimise understanding, recall, and implementation of healthcare information by the patient, and therefore their healthcare experiences and outcomes.

Health literacy scores of participants in Stage 1 were generally low. Participants were fairly typical of SVHM patients, with 42% born outside Australia and 26% speaking a language other than English (i.e., Vietnamese), requiring an interpreter. During the study year, 48% of all patients admitted to the stroke unit at SVHM were born in a country other than Australia (compared with 31% nationally) and 18% spoke languages other than English (compared with 8% nationally) [32]. Despite most participants being more than six months post-stroke, health literacy scores were lowest for “having sufficient information to manage my health” and “navigating the healthcare system”. This is consistent with previous research where stroke survivors and care givers report receiving an inadequate amount of information, or receiving information at an inappropriate time, or stroke survivors not recalling information provided [33].

These findings reinforce the need for clinicians to consider all the factors that impact healthcare communication, including health literacy, preferred language, older age, cognition, aphasia and other communication disability, and available support when developing and delivering health information. The subset of participants who completed Stage 2, several of whom had lived experience of these issues, were able to provide simple practical recommendations for clinicians to support people with such challenges (listed in Table 3). They suggested comprehensive consideration of patient characteristics that are likely to impact understanding of and engagement with health information (to address the issues raised in Theme 1); tailoring of information using communication support strategies such as the use of pictures, videos, and gesture to support text and spoken language; and providing notes, handouts, and resource links for patients to refer to outside of clinical appointments (Theme 2). These recommendations are consistent with strategies used in evidence-based cognitive rehabilitation programs [34,35], aphasia-friendly written education materials [36,37], and healthcare communication post-stroke [38].

Efforts to tailor information to patients’ individual health literacy require clinicians to assess and understand, rather than assume, their patients’ prior knowledge, capacity, and beliefs about stroke and health services. Assessment of health literacy goes beyond standard stroke assessment tools, but can be achieved fairly quickly by asking questions such as “What do you understand about stroke?” Similarly, assessment of cognitive and communication support needs is often overlooked in standard stroke care [39,40] and is not captured by commonly used tools that assess functional outcomes such as the modified Rankin scale [41]. In our attempts to purposively recruit participants with cognitive and communication impairments for this study, we found that mRS scores were not helpful indicators of these difficulties, and there was typically limited identification of cognitive or communication issues in health records. This suggests that indicators of challenges to health literacy may be “flying under the radar”, precluding the opportunity to adequately tailor healthcare communication.

Medical, nursing, and allied health professionals also require training and competencies in how to adapt their communication for people with cognitive impairment [35] and aphasia [42]. Examples of free training include Stroke Foundation [43] and Aphasia Institute [44]. This is often overlooked in training programs [38]. Simply presenting standard information is not sufficient for patients to understand and implement it. Developing clear competency frameworks and associated training protocols to support communication of health information for people with low health literacy is a priority for future research. These protocols should also encourage clinicians to develop a communication style that engenders trust and confidence in their patients. “Feeling like they are in safe hands” (Theme 4) was considered critical by our participants for supporting their engagement with health information and feeling safe to ask questions.

Given the importance of support networks in facilitating access to healthcare information (as reflected in Theme 3), stroke survivors who do not have family members or close others available to support them require clinicians to take extra steps to ensure they understand and feel safe. The subtheme we identified on the *Systemic and societal context influencing individual stroke literacy* also highlights that improving stroke literacy and reducing stigma in the broader community is likely to positively impact people’s experiences of stroke. This points to the importance of community awareness initiatives, not just about early stroke signs (e.g., F.A.S.T) but also campaigns that include information about life after stroke, especially “invisible” difficulties such as fatigue, cognitive impairment, depression, and anxiety, which are commonly overlooked and misunderstood [39,40]. Including people with lived experience as partners in teams dedicated to health service design and public health campaigns can be an effective strategy that enables the needs of people with stroke to be considered.

### Limitations

Our findings should be considered in the context of several limitations. Firstly, there was almost 50% participant attrition between Stage 1 and 2 of the project, particularly of people born outside Australia who spoke Vietnamese and required an interpreter. Of interest, a larger proportion of people who completed both stages were noted to have cognitive impairment than those who did not complete Stage 2 (40% vs. 22%), but fewer had communication impairment (30% vs. 44%), though numbers were small. There were several stated reasons for not participating in Stage 2, but these did not always reflect the extra challenge involved in participating in research for those who do not speak English, which may have formed a barrier to participation even if not directly stated. Notably, Vietnamese-speaking participants tended to prefer face-to-face interviews, but this was not an option throughout a substantial proportion of the data collection period due to pandemic-related restrictions. This meant that while participants from CALD backgrounds were well represented in Stage 1 (and therefore in the case vignettes), they were not as well represented in the focus groups and interviews, so their perspectives on the delivery of health information for the characters in those vignettes were not as prominent. While we deemed theoretical sufficiency to have been reached after 10 interviews, it remains possible that richer information could be gleaned with a more diverse sample. Future research should address this issue, which should be more feasible outside the context of restrictions preventing face-to-face appointments. Additionally, participants were recruited from one stroke unit in Melbourne, Australia. Perspectives of stroke survivors from other regions of Australia and from different CALD communities around the world would be valuable to include in future research.

While there were concerted efforts to recruit people with cognitive and communication impairments, we hoped to recruit more. These efforts were hampered by the lack of information about cognitive and language abilities in health records. Cognitive and language screening, with clear reporting of any impairments in health records, would be a helpful addition to standard stroke care. We originally planned to conduct formal cognitive assessment with participants; however, this plan was revised in the context of pandemic-related lockdowns, given the challenges of assessing cognition via telehealth. While various sources of information were used to identify possible cognitive impairments, the lack of a consistent formal assessment of cognition limited our understanding of participants’ cognitive capabilities and thus the generalisability of results. Future research should incorporate cognitive and language screening to facilitate purposive recruitment as well as characterise samples and ensure inclusion of people with cognitive and communication impairments in stroke research [16]. At a minimum, screening of self-reported cognitive status (thinking and memory) and its impact on daily life is feasible. The extent and impact of language difficulties can be rated by trained examiners using AusTOMs ratings.

## 5. Conclusions

Our study provides valuable perspectives from stroke survivors about the impact of health literacy on their needs and preferences about the delivery of health information. Listening to the voices of those who are typically under-represented has allowed us to identify key improvements that would improve stroke service delivery. These include considering health literacy levels of patient populations when developing stroke information resources, and assessment of individuals’ cognitive and communication support needs as standard practice. Training is necessary to support clinician and researcher competencies in tailored healthcare communication. Community awareness initiatives are recommended to improve stroke literacy in the general population. These improvements are feasible—for example, in response to our findings, the stroke team at SVHM ran clinician workshops on health literacy and have adjusted their service model to ensure responsivity to low health literacy. It will be important to evaluate the impact of these and similar other service improvements. Greater awareness of and response to health literacy in stroke services has the potential to improve patient experiences and outcomes of stroke care.

## Figures and Tables

**Figure 1 healthcare-13-00541-f001:**
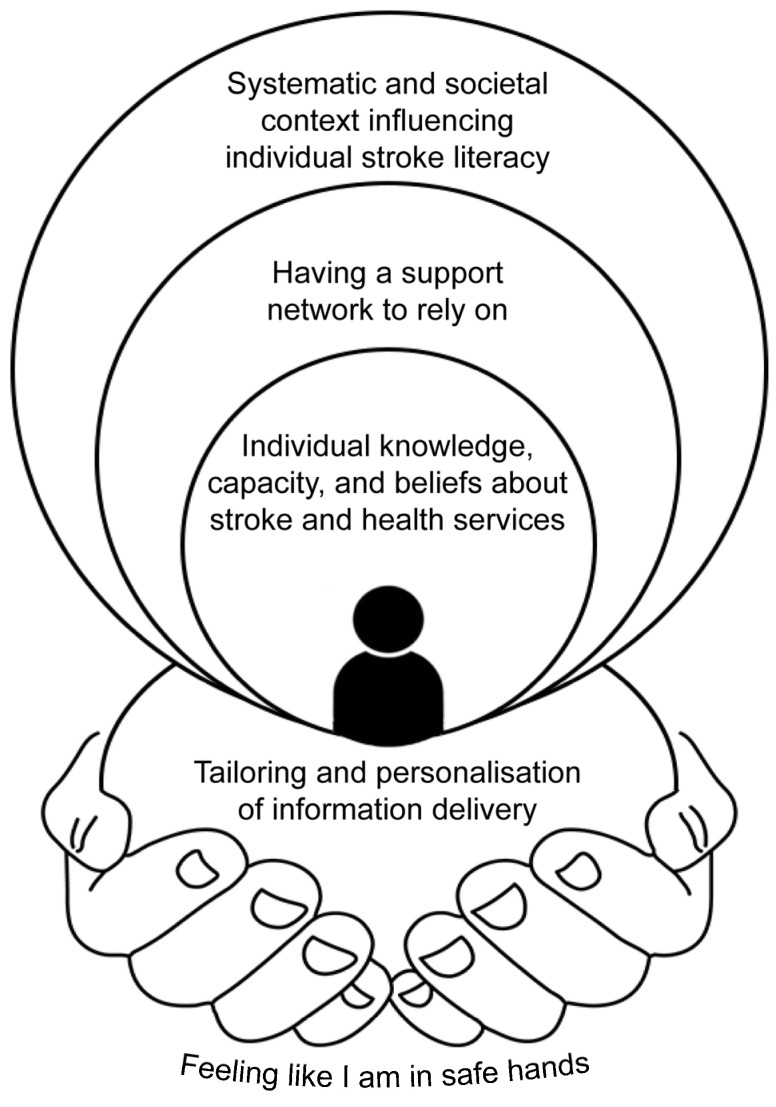
Schematic of the themes and relationship between them.

**Table 1 healthcare-13-00541-t001:** Case vignette example: Character “Mai”.

Description
Mai is a 75-year-old woman who moved to Australia in the 1970s. She was born in Vietnam and only speaks Vietnamese. She did not complete high school and worked in a factory for most of her life. Mai’s husband died two years ago. Her main support is now her daughter who lives close by but works long hours and has two small children. Because of the COVID-19 pandemic, Mai was unable to have visitors while she was in hospital after her stroke. She found this very scary and a lot of the time was unsure what was happening. Her daughter spoke to the doctors each day, but she is not sure what they spoke about. None of the doctors or nurses could speak Vietnamese, nor was she offered an interpreter, so she was unable to ask any questions. The Vietnamese language leaflets she was given did not always make sense to her. Mai has a good GP who speaks Vietnamese. This helps Mai trust him in discussing her problems. Her daughter cannot always come to Mai’s specialist appointments with her. She likes appointments where she has an interpreter—this means she can ask questions. She prefers to hear spoken information rather than have it in writing, as she is not a confident reader. It also gives her chance to socialise. Since having her stroke, she does not very often see people from her community. Mai takes lots of medications and had lots of tests after her stroke. She is not sure what they are all for, but the doctor told her they are important. Her doctors have told her to exercise more and change her diet. Mai likes the food she cooks and walks round the block every other day. She does not think she needs to change this, as she is taking the tablets the doctor told her to.

**Table 2 healthcare-13-00541-t002:** Participant characteristics.

ID	Age Bracket, Gender, Born in/Outside Australia ^a^	Highest Education	SES ^b^	Months Since Stroke	Stroke Type (Oxford)	Stroke Severity (NIHSS) ^c^	mRS ^d^	Communication/Cognitive Impairment ^e^
1 *	60–69, M, Australia	Degree	5	9	Right PACI	5	1	0
2 *	60–69, M, Outside Australia	High school	4	6	Right TACI	5	2	NIHSS 1—mild aphasia; cognitive impairment
3 *	20–29, F, Australia	Certificate/apprenticeship/diploma	2	10	Right POCI	1	1	0
4	60–69, M, Outside Australia	Certificate/apprenticeship/diploma	2	5	Right PACI	1	1	NIHSS 1—dysarthria
5 *	40–49, F, Australia	Degree	5	8	Right PACI with haemorrhagic transformation	2	0	0
6 *	70–79, F, Australia	Primary school	5	14	Left haemorrhage	2	2	Cognitive impairment
7 *	50–59, M, Australia	Post-graduate	5	16	Left POCI	1	1	0
8	50–59, M, Australia	High school	5	15	Left PACI	1	2	0
9*	60–69, F, Outside Australia	Degree	3	8	Left POCI	1	2	0
10	60–69, F, Australia	High school	5	4	Left PACI	3	1	0
11 *	40–49, F, Australia	Certificate/apprenticeship/diploma	1	4	Left TACI	1	1	NIHSS 1—mild aphasia; Cognitive impairment
12	70–79, M, Outside Australia	Primary school	5	9	Bilateral POCI	1	1	Cognitive impairment
13 *	60–69, F, Outside Australia	Degree	1	12	Left POCI	5	1	Cognitive impairment
14	80–89, M, Outside Australia	Primary school	5	11	Left POCI	5	2	Cognitive impairment
15	60–69, F, Outside Australia	Primary school	5	8	Right PACI	5	1	0
16	60–69, M, Australia	Primary school	2	10	Left and right POCI	1	3	NIHSS 1—aphasia
17 *	30–39, M, Australia	Primary school	4	61	Left haemorrhage	5	3	NIHSS 2—aphasia
18	40–49, F, Australia	Degree	5	6	Left haemorrhage	1	2	NIHSS 3—aphasia
19	70–79, F, Outside Australia	Primary school	5	14	Right LACI	5	4	NIHSS 1—mild aphasia

Note: * Completed Stage 1 and 2 of research project; ^a^ demographic information is reported broadly (age bracket, born in/outside Australia) to preserve participant confidentiality; ^b^ SES = socio-economic status (SES) quintile based on postcode of home address; ^c^ National Institutes of Health Stroke Scale (NIHSS) at time of hospital admission, scored by researcher from medical record review; ^d^ mRS = modified Rankin score, established at time of research interview; ^e^ communication and cognition impairments were classified from medical record review and clinical impression of participant during interview, by experienced stroke researcher (CF), and cognitive impairment reported by the family was also noted for participants who were interviewed with an interpreter. LACI = lacunar infarct. TACI = total anterior circulation infarct. PACI = partial anterior circulation infarct. POCI = posterior circulation infarct

**Table 3 healthcare-13-00541-t003:** Health literacy scores for the sample.

Health Literacy Scale	Mean (Standard Deviation)
HLQ scale 2: Having sufficient information to manage my health	2.67 (0.76)
HLQ scale 3: Actively managing my health	3.05 (0.72)
HLQ scale 4: Social support for health	3.24 (0.80)
HLQ scale 6: Ability to actively engage with healthcare providers	3.53 (1.02)
HLQ scale 7: Navigating the healthcare system	3.38 (1.01)
BRIEF Health Literacy Screener; median (inter-quartile range)	12 (8, 19)

**Table 4 healthcare-13-00541-t004:** Cluster analysis, reporting HLQ and BRIEF scores only.

Cluster Number	% of Sample	Having Sufficient Information	Actively Managing Health	Social Support for Health	Active Engages with Health Providers	Navigating Health Services	BRIEF HLST Score
1	47%	3.19	3.49	3.73	3.80	3.76	17.0
This cluster represents nearly half the sample. People in this cluster have reasonably good health literacy overall, but are not completely confident they know all they need to about what happens next after the stroke. Because of limited experience with the health system prior to their stroke, they find it hard to know what services are available, and also what questions they should ask the specialist—this may be because they lack confidence to discuss concerns with them.							
2	5%	1.00	1.40	1.00	5.00	5.00	13.0
This was the smallest cluster in the sample. People in this cluster tend to have little support from family or friends to help with their health, and have large gaps in their knowledge, which may make it difficult to set goals, make plans, and work out how to look after themselves post-stroke. They have no problems talking with providers and being quite assertive in that relationship, and are also able to advocate for themselves in relation to obtaining the right healthcare. However, their BRIEF score is low, which means they may struggle to understand patient education materials, possibly contributing to a perception that they do not have enough information about their condition.							
3	21%	1.94	3.05	3.10	4.05	3.88	7.8
People in this cluster have very large gaps in their knowledge, and feel they are lacking information about their condition. They are quite keen to take responsibility for their health, so providing information that they understand may be very helpful. Their BRIEF score is very low, so it may be that the information they receive is not comprehensible for them. They have quite good social support for health. They also feel reasonably confident in talking with providers and asking questions, and do have some understanding of the health services available to them.							
4	26%	2.65	2.60	2.92	2.32	1.97	20.0
People in this cluster have gaps in their knowledge and do not feel the information they are given is right for them, which may impact on their ability and motivation to manage their health (their “engagement”). They report only moderate social support for health. People in this cluster are very passive in their interaction with health providers and have limited knowledge of how the healthcare system works and where to find the right providers.							

**Table 5 healthcare-13-00541-t005:** Descriptions of key concepts and illustrative quotes for Theme 1, “Individual knowledge, capacity, and beliefs about stroke and health” and its subtheme, “Systemic and societal context influencing individual stroke literacy”.

Concept Reflected in Theme	Description	Quotes
Prior knowledge of stroke	Prior knowledge of stroke and experience navigating healthcare services helped participants to understand what was happening to them, “how things work” and to access appropriate support. Not having prior knowledge about stroke made it more difficult for participants to understand health information.	*Doctors just assume that you know [there are a range of stroke outcomes]. I felt very confused when I was in hospital because of that… I didn’t know the repercussions of a stroke. I didn’t know what happens afterwards. I didn’t know any of that*.—P9
Individual capacity to engage with healthcare information	Individual capacity was central to how participants understood, recalled, and implemented healthcare information. Capacity was affected by stroke-related characteristics such as changes to cognitive function (memory, attention, planning) and motor ability, as well as demographic factors such as language spoken at home. Emotional responses to stroke such as frustration and feeling scared and incapable also impacted participants’ capacity to advocate for themselves and engage with healthcare information. For instance, some participants reported that fear of appearing “stupid” made them feel less comfortable asking questions of their healthcare provider.	*[responding to case vignette] She already feels a bit worried about asking questions that might be silly, and that will put her on the back foot a little bit more.—*P5
Psychological factors	Individual psychological factors which participants considered to enhance understanding and implementation of healthcare information included high levels of confidence, optimism, tolerance of uncertainty, and being in a state of readiness to change.	*[responding to case vignette] He’s been told over and over a thousand times [to make lifestyle changes], but unless you acknowledge it and are ready, it’s not going to make any difference*.—P11
Subtheme: Systemic and societal context influencing individual stroke literacy	This subtheme highlights the influence of prevailing social and cultural narratives on individual knowledge, capacity, and beliefs. These narratives appeared to influence participants’ experience of stroke, their stroke literacy, and how actively they sought support. Several participants described stigma and mystery surrounding stroke, which influenced how easily they understood information presented to them, and how they responded to the signs of their own stroke.	*I didn’t know I had stroke. I didn’t go to the doctor for four days afterwards because no one told me that this was a stroke.—*P9
Stigma and mystery around stroke	Stigma and mystery surrounding stroke also influenced participants’ expectations for their post-stroke lives. Several participants mentioned that prior to their stroke they held the belief that people “become a vegetable” following a stroke. Having an expectation that people who have had a stroke are unable to be helped, are not worth helping, or will be given up on by society was thought to influence patient engagement with healthcare information.	*Sometimes people think that having a stroke is like, it’s the end of the world. It’s not, but it makes people feel like that’s it. You become a vegetable, you can’t do anything for yourself.—*P9

**Table 6 healthcare-13-00541-t006:** Descriptions of key concepts and illustrative quotes for Theme 2, “*Tailoring and personalisation of information delivery*”.

Concept Reflected in Theme	Description	Quotes
Feeling overwhelmed and confused	Many participants reported feeling confused by the way information was presented to them, or overwhelmed by too much information at once.	*When the word is too big, it’s just too hard.—*P17 *Sometimes when you’re given information, you get confused, it jumbles up in your brain…you feel like an idiot because, oh, you should know that, but it’s just confused.—*P9
The value of accessible information delivery	A frequent suggestion was to supplement spoken information with visual information (including using videoconference rather than telephone for telehealth interactions), experiential learning, and to offer accessible take-home materials.	*It sometimes depends on what’s being learned, but sometimes it’s easy to do things in real life and being shown things in real life that you’re trying to do… or you need someone there with you to explain it first for something that you’re looking at for the first ever time.—*P5 *The simpler, the message the better…We don’t need to know all the technicalities of stroke, we just need to know what to do and what [are] the impacts*.—P1
The value of interpreters and translations	For participants who did not understand English well, translators and translated materials were essential to understanding healthcare information.	P13: *I can’t read English.* Interviewer: *I hear that they gave you documents in English, which is very unhelpful because you can’t read it.* P13: *Yes*.
Having resources available and explained	Participants appreciated having standard resources such as the My Stroke Journey pack. Many participants identified that it was helpful for a clinician to step them through the resource so that they knew which parts were relevant and how they might apply that information in their lives.	*The human experience of learning is important rather than just being given things to read.—*P7
Clinician support and strategies to enhance understanding and recall	Participants used a variety of strategies to enhance their understanding and recall, including taking notes and recording appointments, and using the Internet to supplement supplied information between appointments. However, not all participants had the capacity to use such strategies following their stroke. For instance, one participant had difficulty using their hand to write. Proactive behaviour by the healthcare provider, such as checking patient comprehension and providing take-home written information and resources, was appreciated by participants. Active, regular monitoring and being sent reminders about upcoming appointments were also identified as assisting with recall and implementation of health information.	*I’m supposed to get my bloods done every six months. I don’t like going…but [the GP] will nag me until I go.—*P9
Obtaining the information that is needed	Participants also highlighted the importance of tailoring information to their specific circumstances and knowledge. This included assisting them to identify achievable goals, and taking into account their primary concerns, their understanding of what happened to them, and other important aspects of their world. For instance, participants commonly identified the prevention of future stroke as a primary concern. Not having sufficient understanding of the cause of their stroke caused considerable worry. In some instances, this worry undermined healthcare information. One participant expressed anxiety about advice to return to exercise, knowing that high blood pressure had contributed to their stroke.	*I was exercising when it happened. If I exercised to that point, is it going to happen again? Was there something that I was doing that made that happen?—*P5

**Table 7 healthcare-13-00541-t007:** Recommendations from participants about tailoring and personalising delivery of information for people with stroke.

**Patient Characteristics to Consider**
Are there motor and mobility challenges (including writing)?
2. Are there speech and communication difficulties?
3. Are there cognitive difficulties including attention, planning, and memory?
4. What is their language ability?
5. What is their prior knowledge?
6. Who is the audience? The patient? Their support people?
7. What support do they have to engage with the information? Family? Friends? Other support?
8. What information is most important at this stage in their journey?
**Information Delivery and Resources**
Tailor information to patient concerns and goals
2. Include practical information to inform day-to-day decisions
3. Encourage questions
4. Ask the patient to summarise what has been discussed to gauge understanding (teach-back)
5. Make translations and translators available
6. Turn on video for telehealth calls and support information delivery with visual cues
7. Be proactive in offering to take notes and provide take-home materials
8. Deliver the information in a number of formats, e.g., talking and writing notes
9. Keep written information simple; avoid jargon, use plain English, consider resources in Easy English
10. In written materials, highlight the main points with good design and use dot points
11. Use diagrams and pictures to supplement written information
12. Consider sharing document summaries or ”fact sheets” that highlight the key messages
13. Offer audio and video take-home materials, not just written materials
14. Offer high-quality, reliable Internet resources (that are flagged as being from a trustworthy source)

**Table 8 healthcare-13-00541-t008:** Descriptions of key concepts and illustrative quotes for Theme 3, “*Having a support network to rely on*”.

Concept Reflected in Theme	Description	Quotes
Family and friends as interpreters/translators and advocates	Supportive friends and family assisted in interpreting information for participants in a way they could understand. In some instances, friends and family translated information into the participant’s preferred language. Family and friends played an important advocacy role for participants within healthcare settings, helping the participants’ needs and preferences to be heard. Supportive friends and family assisted in managing healthcare concerns, such as treatment adherence, and ensuring participants attended appointments.	*When the doctors tell me what to do I can’t understand…I’ll try to listen and try take in everything they say, but I can’t do that… but by my friend to explain it a little different, then I get it.—*P17
Lacking access to close others is a barrier	Not having access to supportive family and friends was described as a major barrier to navigating the complexity of the healthcare system, understanding, recalling and implementing healthcare information, and feeling supported in recovery. One participant noted that not being able to take support people to appointments due to COVID-19 restrictions was difficult. Another participant noted that living in a different city to their family negatively impacted their sense of agency, access to services, and ability to implement healthcare advice.	*At the time when my blood pressure become very high, then I find that difficult because I’m on my own.*—P13
Having access to different kinds of support	Several participants stressed that it was important for patients to know they can bring support people to appointments, and that when family or friends were not available to support and advocate for patients, a professional advocate could fill this role. Accessing peer support from other stroke survivors was helpful for some participants. These participants found it beneficial to engage with people who understood their experiences and could offer practical guidance to navigate their new post-stroke reality.	*If you got someone, you feel much comfortable and much better off than on your own.—*P2
Feeling isolated versus connected	Several participants mentioned that friendships and acquaintances dropped away following their stroke, perhaps because of stigma surrounding stroke and low stroke literacy in the community. Feeling connected with others gave a sense of being part of a support network. The Stroke Foundation ”Enable Me” newsletter, which contains stories and advice about life after stroke, was highlighted as being useful and reaffirming.	*After you’ve had a stroke… I found that people don’t want to talk to you about it…I don’t think it’s [that] they don’t want to know. I think they just don’t know what to say.*—P9 *Every week they email you a newsletter. It does make a difference actually because you see other people’s stories in there and you don’t feel alone.—*P9

**Table 9 healthcare-13-00541-t009:** Descriptions of key concepts and illustrative quotes for Theme 4, “*Feeling like I am in safe hands*”.

Concept Reflected in Theme	Description	Quotes
Feeling safe to speak up and ask questions	Participants described the importance of feeling like they could ask questions and clarify information if they did not understand.	*If you are seeing a doctor that you feel safe and comfortable with… you’re more likely to speak up and say that you don’t understand. Whereas, if it’s a doctor that comes in… speaking very fluently in their medical terminology, you might go, “Okay, yes, thank you,” and just walk out of there with no idea. If you feel safe and comfortable, you might be more likely to say, “I have no idea what you said”.—*P11
Person-centred care is crucial	Being seen as a person rather than a condition was of central importance to participants. Having a caring, collaborative relationship that centred patient goals, strengths, and capacities enhanced the feeling of being seen as a “whole person”. Features of collaborative relationships included being included in decision making, creating a safe environment, not assuming prior knowledge of stroke, and having sufficient time to ensure patients understand information and can ask questions. This also helped to build confidence in the treating team, foster trust in the information supplied, and enhanced the feeling of being in safe hands.	*At the time when I was in hospital… I actually felt like people were making decisions around me and not including me.—*P9
Not feeling rushed	Many participants reported that interactions with healthcare providers felt rushed. Feeling rushed discouraged participants from asking questions and exacerbated stroke-related cognitive difficulties.	*I feel sometimes rushed… When things are rushed…it becomes anxious and then you forget everything.—*P6
Coordinated care and consistent messages	Participants reported feeling in safe hands when their care was well coordinated. This included how well individual practitioners appeared to coordinate patient information and progress, such as following up on test results, making monitoring appointments, and making referrals to additional services. It also included the coordination of the broader healthcare team in sharing reports and test results where appropriate, and providing consistent messaging to the patient. Participants reported being particularly confused when multiple members of a team gave them different information during their hospital stay.	*All the medical terminology and getting information from this person, that person, and just information overload.—*P11
Having access to appropriate healthcare	Being linked in with suitable healthcare professionals was also an important part of feeling in safe hands. Participants who lived regionally, who did not have strong social support, or who spoke a language other than English had difficulty accessing appropriate medical and support services.	*Because I’m rural…when I went to ED [Emergency Department] with my stroke, I was left in ED for six hours before I even got a bed… [in that time] I could have driven to Melbourne and potentially been getting treated.—*P11
Having a trusted point of contact	Participants appreciated help accessing additional support such as from the Stroke Foundation, and acknowledged the important role of GPs and social workers in helping to make these connections. Some participants suggested that having a trusted point of contact they could get in touch with to ask questions between appointments would be useful and reassuring.	*I had help from a social worker. That was the best thing because he made me aware of the services that were around. If they hadn’t done that, I’d still be sitting here thinking, “Well, I don’t know how to do this*”.—P9

## Data Availability

Data can be made available upon reasonable request to the authors.

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
