# Peer review of "“When the Word Is Too Big, It’s Just Too Hard”: Stroke Survivors’ Perspectives About Health Literacy and Delivery of Health Informationâ€"

_healthcare, 2025, doi:10.3390/healthcare13050541_

Round 1
Reviewer 1 Report
Comments and Suggestions for Authors
Dear Authors,
Thank you for providing me with the opportunity to review this interesting paper. Below, I have listed my comments:
1) The introduction includes some important statistics (e.g. only 41% of Australian adults have sufficient health literacy), but it could strengthen the connection between these data points and the subsequent discussion. For instance, after stating the low prevalence of adequate health literacy, you could more explicitly highlight how this specifically impacts stroke survivors, providing a stronger rationale for the study.
2) While the concept of organizational literacy is introduced, it could be elaborated on a bit more to make it clearer for readers unfamiliar with the term. For instance, explaining how poor organizational literacy specifically affects patient outcomes, and perhaps giving examples, would make the argument even stronger.
3) The focus group process is described, but it could benefit from more details on how the interviews were moderated. For instance, how did the team ensure that participants were encouraged to share their views freely? A brief mention of how participants were encouraged to speak or how the interviewers ensured an inclusive environment would be helpful.
4) It’s mentioned that a formal cognitive assessment was planned but not conducted due to challenges with telehealth. While this is understandable, it could be worthwhile to briefly discuss any potential impact this may have had on the study's findings or limitations.
5) In the discussion section, while this part presents findings and then gives recommendations, it could be more effective if these were more closely integrated. For example, after discussing each theme (like Tailoring and personalisation of information delivery), the corresponding recommendation (such as the use of videos or handouts) could immediately follow to reinforce the direct application of the findings.
6) While the section discusses the direct implications for clinicians and stroke survivors, it could benefit from a brief mention of the broader implications for policy, healthcare systems, or community-level initiatives. For example, it would be valuable to suggest how the findings could influence the design of stroke rehabilitation programs or public health campaigns.
I hope this feedback is helpful.
Author Response
Thank you very much for taking the time to review this manuscript. Please find the detailed responses below and the corresponding revisions/corrections in track changes in the re-submitted files. Please note that line numbers refer to the manuscript that contains tracked changes.
|
Point-by-point response to Comments and Suggestions for Authors |
Comments 1: The introduction includes some important statistics (e.g. only 41% of Australian adults have sufficient health literacy), but it could strengthen the connection between these data points and the subsequent discussion. For instance, after stating the low prevalence of adequate health literacy, you could more explicitly highlight how this specifically impacts stroke survivors, providing a stronger rationale for the study.
Response 1: Thank you. We have added detail about the prevalence of post stroke cognitive and communication impairments to strengthen the connection between data points, and improved the linking sentence between paragraphs 1 and 2.
Lines 49 to 56 now state “One in two people with stroke experience cognitive impairment and one-in-four have communication impairments (Weterings et al). This can further compromise health literacy and affect comprehension, recall, and effective implementation of health-related information (Liu)). With low health literacy impacting poorer stroke outcomes including medication adherence, general health status and hospital readmission (Bushnell et al., 2014), it is incumbent on health practitioners to provide appropriate health information to support stroke recovery”.
Comment 2: While the concept of organizational literacy is introduced, it could be elaborated on a bit more to make it clearer for readers unfamiliar with the term. For instance, explaining how poor organizational literacy specifically affects patient outcomes, and perhaps giving examples, would make the argument even stronger.
Response 2: Thank you. We have added further information about organisational literacy and that the onus of clear communication is with health practitioners, rather than patients (lines 55-56). Another reference (Coleman et al 2023) has been added for further information about organisational literacy.
Lines 59 to 65 now read “Proponents of improving organisational literacy, rather than attempting to improve health literacy support a “universal precautions” approach to heath communication. Such an approach requires habitual use of clear communications to improve accessibility of health information for all people with stroke, regardless of their health literacy (Coleman et al 2023). This shifts the onus of responsibility from the marginalised patient to that of the health service to provide appropriately tailored stroke care (Hart et al., 2025).”
Comment 3: The focus group process is described, but it could benefit from more details on how the interviews were moderated. For instance, how did the team ensure that participants were encouraged to share their views freely? A brief mention of how participants were encouraged to speak or how the interviewers ensured an inclusive environment would be helpful.
Response 3: Thank you . We have added further information into page 5:
- Line 179 “Participants were also provided with the opportunity to introduce themselves.”
- Line 188 – 192 “The group facilitators encouraged all participants to share their views and ensured that everyone had the chance to do so either verbally or in the Zoom chat. Communication support strategies, such as slowed pace of speech, repetition, paraphrasing, and reflective summaries of participants’ comments to confirm their meaning, were used by facilitators as required”
Comment 4: It’s mentioned that a formal cognitive assessment was planned but not conducted due to challenges with telehealth. While this is understandable, it could be worthwhile to briefly discuss any potential impact this may have had on the study's findings or limitations.
Response 4: Thank you. We added detail into the limitations section (lines 476-87) about how the lack of being able to fully characterise participants’ cognition limits our knowledge of the generalisability of results.
The section now reads “We originally planned to conduct formal cognitive screening assessment with participants, however this plan was revised in the context of pandemic-related lockdowns given the challenges of assessing cognition via telehealth. While various sources of information were used to identify possible cognitive impairments, the lack of a consistent formal assessment of cognition limited our understanding of participants’ cognitive capabilities and thus the generalizability of results. Future research should incorporate cognitive and language screening to facilitate purposive recruitment as well as characterise samples and ensure inclusion of people with cognitive and communication impairments in stroke research (Shiggins et al., 2022). At a minimum, screening of self-reported cognitive status (thinking and memory) and its impact on daily life is feasible. The extent and impact of language difficulties can be rated by trained examiners using AusTOMs ratings.
Comment 5: In the discussion section, while this part presents findings and then gives recommendations, it could be more effective if these were more closely integrated. For example, after discussing each theme (like Tailoring and personalisation of information delivery), the corresponding recommendation (such as the use of videos or handouts) could immediately follow to reinforce the direct application of the findings.
Response 5: Thank you for this suggestion. To better link the themes with concrete recommendations from participants, relevant themes are mentioned throughout the discussion when outlining recommendations. For example lines 402 - 407 now reads “They suggested comprehensive consideration of patient characteristics that are likely to impact understanding of and engagement with health information (to address the issues raised in Theme 1); tailoring of information using communication support strategies such as the use of pictures, videos and gesture to support text and spoken language; and providing notes, handouts and resource links for patients to refer to outside of clinical appointments (Theme 2).
Comment 6: While the section discusses the direct implications for clinicians and stroke survivors, it could benefit from a brief mention of the broader implications for policy, healthcare systems, or community-level initiatives. For example, it would be valuable to suggest how the findings could influence the design of stroke rehabilitation programs or public health campaigns.
Response 6: Thank you for this suggestion. We have included further details about the broader implications as you suggest, and additional training resources available for clinicians and researchers. Training examples are included in line 425 “Examples of free training include Stroke Foundation [44] and Aphasia Institute [45].”
Line 444 includes the importance of partnering with people who have Lived Experience on project teams “Including people with Lived Experience as partners in teams dedicated to health service design and public health campaigns can be an effective strategy that enables the needs of people with stroke to be considered.”
Conclusion now includes the wording: (Lines 490 – 95): “…These include considering health literacy levels of patient populations when developing stroke information resources, and assessment of individuals’ cognitive and communication support needs as standard practice. Training is necessary to support clinician and researcher competencies in tailored healthcare communication. Community awareness initiatives are recommended to improve stroke literacy in the general population…” |
|
Reviewer 2 Report
Comments and Suggestions for Authors
Thank you so much for this opportunity. Please, see below few comments you might consider or clarify:
Theoretical Saturation:
The manuscript claims thematic saturation was reached, but with such a diverse population (stroke survivors, CALD participants, varying literacy levels, and cognitive impairments), more data would strengthen thematic credibility.
Recommendation: Provide more justification for how saturation was determined, especially given the varied health literacy profiles.
The study aimed to include stroke survivors with cognitive impairments, but no formal cognitive assessments were conducted.
Health literacy challenges might be confounded by undiagnosed cognitive deficits.
Recommendation:
Clearly state how cognitive impairment was determined (self-report, clinician impression, or medical records).
If possible, acknowledge this limitation and suggest future studies use validated cognitive screening tools (e.g., MOCA, MMSE).
Participant Attrition:
The 50% dropout rate between Stage 1 and Stage 2 should be discussed more in the limitations section.
Clarify participant characteristics and attrition in the methods section.
Provide a clearer explanation of cognitive impairment classification
Discuss study limitations in more depth (attrition, cognitive screening, and generalizability).
Best wishes
Author Response
Thank you very much for taking the time to review this manuscript. Please find the detailed responses below and the corresponding revisions/corrections in track changes in the re-submitted files. Please note that line numbers refer to the manuscript that contains tracked changes.
Point-by-point response to Comments and Suggestions for Authors
Comment 1: Theoretical Saturation:
The manuscript claims thematic saturation was reached, but with such a diverse population (stroke survivors, CALD participants, varying literacy levels, and cognitive impairments), more data would strengthen thematic credibility.
Recommendation: Provide more justification for how saturation was determined, especially given the varied health literacy profiles.
We note that we stated that theoretical sufficiency was reached, rather than “saturation”. These are two different concepts, as noted by Braun and Clarke (2019,) who provide a range of arguments against the notion of data saturation in the context of reflexive thematic analysis (which is the qualitative approach we adopted for this study). Theoretical sufficiency is achieved when the researcher has gathered enough data to understand the research question under investigation, as opposed to data saturation which is when researchers claim to have collected data to the point where no new information can be found. We acknowledge that new information may have arisen with further interviews with stroke survivors with diverse characteristics; however, after 10 interviews we deemed that we had collected enough information to sufficiently answer our research question. We have now added a reference after the term “theoretical sufficiency” (p.9); and have added the following sentence to the discussion(p.18).:
While we deemed theoretical sufficiency to have been reached after 10 interviews, it remains possible that richer information could be gleaned with a more diverse sample.
Braun, V., and V. Clarke. 2019. To saturate or not to saturate? Questioning data saturation as
a useful concept for thematic analysis and sample-size rationales. Qualitative Research in
Sport, Exercise & Health 1–16. doi:10.1080/2159676X.2019.1704846.
Comment 2:
The study aimed to include stroke survivors with cognitive impairments, but no formal cognitive assessments were conducted.
Health literacy challenges might be confounded by undiagnosed cognitive deficits.
Recommendation:
Clearly state how cognitive impairment was determined (self-report, clinician impression, or medical records).
If possible, acknowledge this limitation and suggest future studies use validated cognitive screening tools (e.g., MOCA, MMSE).
Response 2:
Thank you. We clarified in methods (line 149) that we collected from the electronic health records any description or assessment of language impairment, communication support needs and cognition. “clinical details of stroke including any description or assessment of language impairment, communication support needs and cognition,…” Also line 153 “Observed communication or cognition impairments were also noted during participant interviews which were conducted by researcher CF, an experienced stroke clinician.” This is also noted on table 1.
We added to the methods section that we had intended to use the cognition assessment tool “Oxford Cognitive Screen (Australian version (OCS-AU)” (line 153)
We added (lines 469 – 478) further distinction between screening and assessment of cognition, some simple questions about self-reported cognition status, and the Austoms language assessment: “We originally planned to conduct formal cognitive assessment with participants, however this plan was revised in the context of pandemic-related lockdowns given the challenges of assessing cognition via telehealth. This limited our understanding of participants’ cognitive capabilities and generalizability of results. Future research should incorporate cognitive and language screening to facilitate purposive recruitment as well as characterise samples and ensure inclusion of people with cognitive and communication impairments in stroke research (Shiggins et al., 2022). At a minimum, screening of self-reported cognitive status (thinking and memory) and its impact on daily life is feasible. The extent and impact of language difficulties can be rated by trained examiners using AusTOMs ratings.”
Comment 3: Participant Attrition:
- The 50% dropout rate between Stage 1 and Stage 2 should be discussed more in the limitations section.
- Clarify participant characteristics and attrition in the methods section.
- Provide a clearer explanation of cognitive impairment classification
- Discuss study limitations in more depth (attrition, cognitive screening, and generalizability).
Response 3
Thank you.
- A comparison of proportions of people with cognitive and communication impairments is included in the limitations section (line 450 - 453) “…Of interest, compared to those who did not complete stage 2, a larger proportion of people who completed both stages were noted to have cognitive impairment (40% v 22%), but fewer had communication impairment (30% v 44%), though numbers were small.”
- We have added detail in results section (line 284) to compare the rates of cognitive and communication impairments between those who did and did not complete both stages. Although numbers were small, a larger proportion of people who completed both stages. “[Of those who completed both stages]…Four people (40%) were identified as having cognitive impairment, and three (30%) had communication impairment. The rates of cognitive and communication impairments noted in the group of people who did not complete stage 2 were 2/9 (22%) and 4/9 (44%) respectively.”
- To clarify the classifications of cognition, we include further details about the sources of information to describe cognitive (and language) limitations. Line 129 - 137 (methods) now states “ Electronic health records were reviewed to collect patient data relating to … clinical details of stroke including any description or assessment of language impairment, communication support needs and cognition, … Observed communication or cognition impairments were also noted during participant interviews which were conducted by researcher CF, an experienced stroke clinician.
- Study limitations have been expanded to mention:
- the proportions of those with and without cognitive or communication impairments (as mentioned above)
- challenges and recommendations regarding screening and assessment of cognition, and unknown impact on generalizability of results. Lines 474 – 85 now read “We originally planned to conduct formal cognitive assessment with participants, however this plan was revised in the context of pandemic-related lockdowns given the challenges of assessing cognition via telehealth. This limited our understanding of participants’ cognitive capabilities and generalizability of results. Future research should incorporate cognitive and language screening to facilitate purposive recruitment as well as characterise samples and ensure inclusion of people with cognitive and communication impairments in stroke research (Shiggins et al., 2022). At a minimum, screening of self-reported cognitive status (thinking and memory) and its impact on daily life is feasible. The extent and impact of language difficulties can be rated by trained examiners using AusTOMs ratings.”
Round 2
Reviewer 1 Report
Comments and Suggestions for Authors
Thank you, Authors, for revising the manuscript.
Reviewer 2 Report
Comments and Suggestions for Authors
Thank you for addressing the comments